# LEARNING TO GENERATE ALL FEASIBLE ACTIONS

## ABSTRACT

Several machine learning (ML) applications are characterized by searching for an optimal solution to a complex task. The search space for this optimal solution is often very large, so large in fact that this optimal solution is often not computable. Part of the problem is that many candidate solutions found via ML are actually infeasible and have to be discarded. Restricting the search space to only the feasible solution candidates simplifies finding an optimal solution for the tasks. Further, the set of feasible solutions could be re-used in multiple problems characterized by different tasks. In particular, we observe that complex tasks can be decomposed into subtasks and corresponding skills. We propose to learn a reusable and transferable skill by training an actor to generate all feasible actions. The trained actor can then propose feasible actions, among which an optimal one can be chosen according to a specific task. The actor is trained by interpreting the feasibility of each action as a target distribution. The training procedure minimizes a divergence of the actor's output distribution to this target. We derive the general optimization target for arbitrary f-divergences using a combination of kernel density estimates, resampling, and importance sampling. We further utilize an auxiliary critic to reduce the interactions with the environment. A preliminary comparison to related strategies shows that our approach learns to visit all the modes in the feasible action space, demonstrating the framework's potential for learning skills that can be used in various downstream tasks.

## 1 INTRODUCTION

Complex tasks can often be decomposed into multiple subtasks, with corresponding skills that solve these subtasks. Learning reusable and transferable skills is an active area of research (Kalashnikov et al. (2021); Chebotar et al. (2021); Deisenroth et al. (2014)). However, given a subtask, learning or even defining the corresponding skill is not straightforward. Consider a robotic scenario where a robot is tasked to grasp an object and handle it in downstream tasks. Different downstream tasks can have different optimal grasps if the object has multiple feasible grasping poses. Therefore, a grasping skill cannot be trained based on optimality definitions of individual tasks. However, a grasping algorithm that learned *all feasible* grasps could support all possible downstream tasks even without explicit knowledge thereof during training. The downstream tasks can then select their respective optimal grasp among the proposed feasible options. Therefore, we consider a skill to be defined by the set of all feasible actions of a subtask.

We propose a novel method to train a generative neural network to generate all feasible actions of a subtask by interacting with an environment. The interaction loop is adopted from Contextual Bandit (CB) (Langford et al. (2008)) and Reinforcement Learning (RL) (Sutton & Barto (2018)): the environment presents a state for which the actor selects an action, which is tested in the environment, yielding either a success or failure outcome. As in CB, we limit ourselves to one-step interactions as opposed to sequential multi-step interactions common in RL. However, we do not minimize regret, typically done in CB. Instead, we optimize the final policy as in RL. Unlike CB and RL, the approach does not aim to find one optimal solution for a given problem but aims to learn all feasible ones.

By interpreting the feasibility of each action given a state as a posterior probability distribution over the actions, a target probability density function (pdf) is defined. The actor is trained to minimize a divergence of its output distribution to this target pdf. The training algorithm in the method proposed can be used with any given f-divergence, including Reverse Kullback-Leibler (RKL), Forward Kullback-Leibler (FKL), and Jensen-Shannon (JS). The possibility to use FKL and JS is instrumental

in visiting all the modes of the posterior distribution, as RKL is known to collapse into a single mode (Jerfel et al. (2021)). The training algorithm presented in this paper uses Kernel Density Estimation (KDE) to estimate the pdf of the actor and Monte Carlo integration with importance sampling to estimate the normalization of the target. The divergences are estimated using samples from a proposal distribution which is a separate KDE based on the original samples of the actor. This resampling step is necessary for convergence, which is discussed in Section 3.3. As interactions with the environment are typically costly, an auxiliary critic network imitating the environment is trained simultaneously. The critic's feasibility estimate of an action is then used to form the target distribution.

The learning algorithm has been tested on a planar robotic grasping problem. We test FKL, RKL, and JS divergences and compare them to implementations of maximum entropy (ME) RL and Generative Adversarial Networks (GANs). Besides accuracy, we measure how many grasping modes, i.e., disconnected regions in the action space, are visited by each approach. Generating actions in all grasping modes can ensure that the learned skill is reusable and transferable for various downstream tasks.

The contributions of this paper are the following:

- Design of a new learning method for generative neural network models to explicitly learn to generate all feasible actions.

- Introduction of a novel gradient estimator for f-divergences that takes advantage of KDEs, resampling, and importance sampling.

- Application of the proposed learning algorithm to a 2D robotic grasping problem, comparing the proposed gradient estimators for f-divergences with related methods.

The rest of this work is structured as follows. Section 2 discusses the related work. Section 3 describes the optimization problem followed by the methodology in Section 4. The evaluation setup is described in Section 5 and the results are presented in Section 6. Section 7 concludes and gives an outlook on future work.

## 2 RELATED WORK

CBs have been successfully applied to several interactive learning problems with discrete action spaces (Langford & Zhang (2007); Agarwal et al. (2014); Foster & Rakhlin (2020); Simchi-Levi & Xu (2021)). In several cases, the context and action spaces have been embedded in a linear multidimensional action space. The embedding keeps the interaction linear while the action and context embeddings can be non-linear (Chernozhukov et al. (2019); Foster et al. (2020); Zhu et al. (2022)). Recently, there has been an increased interest in extending the approach to continuous action spaces. However, most works are limited to 1D actions (Chernozhukov et al. (2019); Majzoubi et al. (2020); Zhu & Mineiro (2022)).

Learning from an interactive environment is also the focus of RL (Sutton & Barto (2018)). Many RL approaches use Actor-Critic (AC) architectures, among which the Soft Actor-Critic (SAC) algorithm (Haarnoja et al. (2018)) is most related to our work. In SAC, the state-action value function of the critic is transformed into an energy-based distribution (Haarnoja et al. (2017)), yielding the target of the actor's distribution. SAC uses RKL as the loss function for the actor, which yields maximum entropy RL. Through a reparameterization trick, which usually uses the family of Gaussians, the RKL is minimized through a direct gradient from the critic.

GANs propose a similar architecture to AC, training a generator and discriminator adversarially. This adversarial training is equivalent to minimizing the JS divergence (Goodfellow et al. (2014)) and has been extended to arbitrary f-divergences (Nowozin et al. (2016)). Conditional GANs (Mirza & Osindero (2014)) offer an alternative solution to the posterior sampling problem, as a generator conditioned on a given state can be trained to provide successful actions adversarially. However, the problem analyzed in our paper is not naturally adversarial, as actions that have not yet been tested in the interactive environment should not be implicitly rejected. The discriminator learns to discriminate between tested successful actions from untested ones, providing the generator with inconsistent gradients.

Expert knowledge is used in Imitation Learning (IL) to derive a policy from demonstration data. The policy may be learned in a supervised manner in behavior cloning (Pomerleau (1988)) or as a combination of Inverse Reinforcement Learning (IRL) (Ng et al. (2000)) to learn a reward function and a subsequent RL procedure. Ho & Ermon (2016) introduced Generative Adversarial Imitation Learning (GAIL), mitigating the intermediate IRL step by using a generative adversary. As discussed in Li et al. (2017), the policy learned by GAIL tends to interpolate between modes leading to erroneous behavior in multimodal settings. Using f-divergence minimization, the authors in Ke et al. (2020); Ghasemipour et al. (2020) intentionally collapse modes to avoid interpolation. IL and adversarial approaches require large amounts of expert data. However, expert data is limited in an interactive environment. Additionally, given that we aim to find all feasible actions, even more expert data representative of all feasible actions would be required.

Posterior sampling has been a long-standing problem in statistics. State-of-the-art methods in Bayesian statistics rely on Markov Chain Monte Carlo (MCMC) algorithms (Hastings (1970); Gelfand & Smith (1990)), eliminating the need to normalize the distribution which is often an intractable problem (Kruschke (2015)). Variational Inference (VI) relies instead on fitting the posterior with a family of parametric probability distributions that can then be sampled from (Jordan et al. (1999); Wainwright & Jordan (2008)). Neural samplers offer another alternative by approximating the posterior with a generative neural network (Nowozin et al. (2016); Hu et al. (2018)). Normalizing flows also infer the pdf for each sample using invertible mappings (Rezende & Mohamed (2015); Tabak & Turner (2013); Tabak & Vanden-Eijnden (2010)). While this approach does not require density estimates, it limits its applicability to specific neural network designs.

For robotic grasping, Kalashnikov et al. (2018) propose using Deep Reinforcement Learning (DRL) to find optimal grasps through interaction with multiple real-world robots. If the goal is to find grasping poses explicitly to be used as the target of a classical controller, supervised learning techniques are often utilized (Kleeberger et al. (2020)). To support various downstream tasks, it would be necessary to find all feasible grasps. To this end, the action space is typically discretized and grasping success is estimated for each discrete action through heat-maps. This can be learned supervised (Kumra et al. (2020); Morrison et al. (2020)) or self-supervised (Zeng et al. (2020)). Zeng et al. (2020) explicitly utilize structure given by spatial equivariances. We aim to find a solution that needs neither discretization nor to make use of the structure as these requirements restrict applicability to planar picking in carefully crafted environments.

## 3 OPTIMIZATION PROBLEM

### 3.1 PROBLEM FORMULATION

An interactive environment, simulated or physical, is defined as a function $g : \mathcal{S} \times \mathcal{A} \mapsto \{0, 1\}$, where $\mathcal{S}$ is the state space of the problem, and $\mathcal{A}$ is the corresponding action space. For all $s \in \mathcal{S}$ we associate a feasible action space $\mathcal{A}_s^+$ such that $g(s, a) = 1, \forall a \in \mathcal{A}_s^+$ and an infeasible action space $\mathcal{A}_s^-$ such that $g(s, a) = 0, \forall a \in \mathcal{A}_s^-$, with $\mathcal{A}_s^+ \cup \mathcal{A}_s^- = \mathcal{A}$. The goal of this work is to find a state-dependent surjective map $\pi_s : \mathcal{Z} \to \mathcal{A}_s^+$, referred to as policy, where $\mathcal{Z}$ is a latent space of appropriate dimensionality. For a given state and uniform sampling from the latent space $\mathcal{Z}$, the pdf of $\pi_s$ is a function $q_s : \mathcal{A} \mapsto \mathbb{R}$, which is the posterior distribution $q_s(a) = q(a|s)$. For the same state, the distribution of the feasible actions according to the environment can be defined as

$$p_s(a) = \frac{g(s, a)}{\int_{\mathcal{A}} g(s, a) \, da}, \tag{1}$$

which is the true posterior $p_s(a) = p(a|s)$. The optimal solution satisfies $\mathcal{D}_f\left(p_s \,\|\, q_s\right) = 0$, where $\mathcal{D}_f$ is an f-divergence, for example from Table 1. This implies $p_s = q_s$, therefore the support of $q_s$ is equal to the support of $p_s$, which is $\mathcal{A}_s^+$ by definition. Thus, the optimal policy is the solution to the optimization problem:

$$\tilde{\pi}_s = \mathrm{argmin}_{\pi_s \sim \Pi} \, \mathcal{D}_f\left(p_s \,\|\, q_s\right), \tag{2}$$

with $\Pi$ being an arbitrary family of distributions. To generalize over all states $s \in \mathcal{S}$, the policy can be modeled as a neural sampler $\pi_\theta : \mathcal{S} \times \mathcal{Z} \mapsto \mathcal{A}$, with a corresponding pdf $q_\theta(a|s)$, where $\theta$ indicates the parameters of the neural network. Assuming that the environment can be used efficiently for repeated controlled experiments, i.e., testing several actions for the same state, the above optimization problem can be solved directly on the environment. If this is not possible, a critic network can be

|  | $f(t)$ | $f'(t)$ |
|---|---|---|
| Jensen-Shannon (JS) | $\frac{1}{2}\left[(t+1)\log\left(\frac{2}{t+1}\right) + t\,\log(t)\right]$ | $\frac{1}{2}\log\left(\frac{2t}{t+1}\right)$ |
| Forward Kullback-Leibler (FKL) | $-\log(t)$ | $-\frac{1}{t}$ |
| Reverse Kullback-Leibler (RKL) | $t\,\log(t)$ | $\log(t)+1$ |

Table 1: Non-exhaustive list of f-divergences and the corresponding first derivative for gradient estimators. The f-divergences are obtained by substituting the $f$ functions above in equation 3 and setting $t = q_\theta/p$. The conventions for $p$, $q$, FKL and RKL assume that $p$ is the target distribution, $q$ is the model, and the FKL divergence is $\int p \log(p/q)$.

used to imitate the environment, which is further discussed in Section 4. Note that the system state is often only partially observable, and the action must be inferred from an observation. For simplicity of notation in the following derivation of the gradients, we assume that the state is directly observable, and we omit the state and action dependence of $q$ and $p$.

## 3.2 f-DIVERGENCES

The f-divergence between two pdfs $p$ and $q$ is a generalization of the Kullback-Leibler (KL) divergence and has the form (Liese & Vajda (2006))

$$\mathcal{D}_f(p\,\|\,q_\theta) = \int_{\mathcal{A}} p\, f\left(\frac{q_\theta}{p}\right)\,da, \tag{3}$$

where $f : (0,\infty) \to \mathbb{R}$ is a convex function. Different choices of $f$ lead to well known divergences as summarized in Table 1. The gradients of the f-divergence w.r.t. $\theta$ can be estimated commuting the derivative with the integral (L'Ecuyer (1995)) and using the score function gradient estimator (Kleijnen & Rubinstein (1996)) as

$$\frac{\partial}{\partial \theta}\mathcal{D}_f = \frac{\partial}{\partial \theta}\int_{\mathcal{A}} p\, f\left(\frac{q_\theta}{p}\right)\,da = \int_{\mathcal{A}} p\, f'\left(\frac{q_\theta}{p}\right)\frac{1}{p}\frac{\partial}{\partial \theta}q_\theta\,da = \int_{\mathcal{A}} q_\theta\, f'\left(\frac{q_\theta}{p}\right)\frac{\partial}{\partial \theta}\log q_\theta\,da, \tag{4}$$

using the fact that $p$ does not depend on $\theta$. Since $q_\theta$ is normalized to 1 and thus $\partial_\theta \int_{\mathcal{A}} q\,da = \int_{\mathcal{A}} q\,\partial_\theta \log q\,da = 0$, a Lagrangian term $\lambda$ can be added to the gradient estimator:

$$\frac{\partial}{\partial \theta}\mathcal{D}_f = \int_{\mathcal{A}} q_\theta\left(f'\left(\frac{q_\theta}{p}\right) + \lambda\right)\frac{\partial}{\partial \theta}\log q_\theta\,da. \tag{5}$$

If the support of $q_\theta$ includes all of $\mathcal{A}$ the above formula can be rewritten as the expectation on $q_\theta$ as

$$\frac{\partial}{\partial \theta}\mathcal{D}_f = \mathbb{E}_{q_\theta}\left[\left(f'\left(\frac{q_\theta}{p}\right) + \lambda\right)\frac{\partial}{\partial \theta}\log q_\theta\right]. \tag{6}$$

Sampling from a proposal distribution $q'$, the expectation can be computed with importance sampling (Robert & Casella (2004); Liu (2001)) as

$$\frac{\partial}{\partial \theta}\mathcal{D}_f \approx \mathbb{E}_{q'}\left[\frac{q_\theta}{q'}\left(f'\left(\frac{q_\theta}{p}\right) + \lambda\right)\frac{\partial}{\partial \theta}\log q_\theta\right]. \tag{7}$$

## 3.3 GRADIENT ESTIMATION

Given a sample $a \sim q_\theta$, it is not possible to directly evaluate $q_\theta(a)$ as it is not available in closed form. Therefore, $q_\theta$ needs to be estimated to compute the gradients of the f-divergence. Given $N$ sampled actions $a_i \sim q_\theta$, $q_\theta$ can be approximated with a KDE by

$$q_\theta(a) \approx \hat{q}_{\theta,\sigma}(a) = \frac{1}{N}\sum_{a_i \sim q_\theta} k_\sigma(a - a_i), \tag{8}$$

where $k_\sigma$ is a Gaussian kernel with a diagonal bandwidth matrix $\sigma$. The KDE makes the estimate of the expectation possible. Using equation 6, computing the expectation value as the average over the samples yields

$$\frac{\partial}{\partial \theta}\mathcal{D}_f \approx \frac{1}{N}\sum_{a_i \sim q_\theta}\left(f'\left(\frac{\hat{q}_{\theta,\sigma}}{p}\right) + \lambda\right)\frac{\partial}{\partial \theta}\log \hat{q}_{\theta,\sigma}. \tag{9}$$

This gradient estimator turned out not to converge in our experiments. While a systematic investigation of the convergence issue was not completed, we suspect two primary reasons for this. First, the support $q_\theta$ usually does not cover the whole action space $\mathcal{A}$, which is necessary for the expectation formulation in equation 6. Second, evaluating $q_\theta(a_i)$ based on a KDE, which uses $a_j$ as supports, has a bias for $j = i$.

Adding Gaussian noise to the samples gives full support in $\mathcal{A}$ and reduces the bias at the support points of the KDE, which lead to convergence in the experiments. The new smoothed samples are given by $a_j^* = a_i + \epsilon$ for $mi \leq j < m(i+1)$ and $\epsilon \sim \mathcal{N}(0, \sigma')$, where $m$ indicates the number of smoothed samples drawn for each original sample. This is equivalent to sampling from a KDE with $a_i$ as supports and $\sigma'$ as bandwidth. The gradient, using importance sampling in equation 7, can be rewritten after resampling as follows

$$\frac{\partial}{\partial \theta} \mathcal{D}_f \approx \frac{1}{M} \sum_{a_j^* \sim \hat{q}_{\theta,\sigma'}} \frac{\hat{q}_{\theta,\sigma}}{\hat{q}_{\theta,\sigma'}} \left( f' \left( \frac{\hat{q}_{\theta,\sigma}}{p} \right) + \lambda \right) \frac{\partial}{\partial \theta} \log \hat{q}_{\theta,\sigma}, \tag{10}$$

with $M = mN$. Additionally, equation 10 requires an estimate of $p$, which in turn requires an estimate of the volume in equation 1

$$\int_{\mathcal{A}} g(a) \, da \approx \frac{1}{M} \sum_{a_j^*} \frac{g(a_j^*)}{\hat{q}_{\theta,\sigma'}(a_j^*)}. \tag{11}$$

This estimation is similar to self-normalized importance sampling (Murphy (2012)) but uses the proposal distribution. The bandwidth $\sigma'$ of the proposal distribution is a hyper-parameter. Setting $\sigma' = c\sigma$, experiments show that $c > 1$ helps convergence. Intuitively, a larger bandwidth enables the exploration of nearby modes in the action space. Specific estimators for the different f-divergences can be obtained substituting $f'$ from Table 1 into equation 10. A summary of the gradient estimators used in this work is given in Table 2.

## 4 METHODOLOGY

The derivation in Section 3.3 assumes that the training could be performed directly on the interactive environment. To train the actor, multiple actions have to be evaluated for the same state. Typically, this is not possible, either because of reproducibility in real experiments or computational cost for simulations. An auxiliary neural network $\xi_\phi : \mathcal{S} \times \mathcal{A} \to \mathbb{R}$ with parameters $\phi$, can be trained to imitate the environment $g$. The policy can then be trained to match the distribution of the feasible actions according to this auxiliary neural network. We refer to $\pi_\theta$ and $\xi_\phi$ as generative actor and critic, respectively. The neural network architectures are presented in Appendix B.

The learning algorithm presented in this paper is inspired by RL and CB. At every training step, the environment generates a state for which the actor proposes an action. The action is evaluated in the environment yielding success or failure. The state, action, and feasibility form an experience stored in a replay memory. The critic is trained on random samples of experiences from the replay memory with a cross-entropy loss on the outcome. The actor trains on a batch of states from the memory. For each state, multiple actions are sampled from the actor, used as support for $\hat{q}_{\theta,\sigma}$ and $\hat{q}_{\theta,\sigma'}$. New samples are drawn from the proposal distribution $\hat{q}_{\theta,\sigma'}$. These samples are evaluated by the critic $\xi_\phi$, and the gradients are computed according to equation 10. The algorithm of the interaction loop can be found in Appendix C. While the general interaction loop is standard in RL, two changes have proven beneficial to the convergence: balanced replay memory and maximum uncertainty collection. Additionally, an action optimization can take advantage of the density estimate to improve performance after training.

### 4.1 BALANCED REPLAY MEMORY

Since the environment yields either a success or a failure, the critic is a binary classifier that suffers from unbalanced data when being trained. Its memory dataset continuously grows through the interactions between the actor and the environment. In the beginning, the actor performs poorly, yielding primarily experiences with failure labels, with the opposite at the end of the training. This labeling bias prevented the critic from distinguishing between success and failure outcomes, making

| Loss | Actor Gradient Estimator | $\lambda$ |
|------|--------------------------|-----------|
| JS | $\frac{1}{2M} \sum_{a_j^*} \frac{\hat{q}_{\theta,\sigma}}{\hat{q}_{\theta,\sigma'}} \log\left(\frac{2\hat{q}_{\theta,\sigma}}{p+\hat{q}_{\theta,\sigma}}\right) \frac{\partial}{\partial\theta} \log \hat{q}_{\theta,\sigma}$ | 0 |
| FKL | $-\frac{1}{M} \sum_{a_j^*} \frac{p}{\hat{q}_{\theta,\sigma'}} \frac{\partial}{\partial\theta} \log \hat{q}_{\theta,\sigma}$ | 0 |
| RKL | $\frac{1}{M} \sum_{a_j^*} \frac{\hat{q}_{\theta,\sigma}}{\hat{q}_{\theta,\sigma'}} \log\left(\frac{\hat{q}_{\theta,\sigma}}{p}\right) \frac{\partial}{\partial\theta} \log \hat{q}_{\theta,\sigma}$ | -1 |
| GAN | $\frac{1}{N} \sum_{a_i} \frac{\partial}{\partial a} \log(1-\xi_\phi) \frac{\partial}{\partial\theta} a_i$ | - |
| ME | $\frac{1}{N} \sum_{a_i} \frac{\partial}{\partial\theta} \log \hat{q}_{\theta,\sigma} - \frac{\partial}{\partial a} \log \xi_\phi \frac{\partial}{\partial\theta} a_i$ | - |

Table 2: Gradient estimators of different losses and choice of Lagrangian multiplier $\lambda$.

convergence impossible. To avoid the critic from biasing towards failure or success labels, we use two replay memories, one for failures and one for successes. When training the critic, half of the experiences are sampled from the positive replay memory and the other half from the negative replay memory. With this strategy, the labeling bias can be mitigated. The potentially amplified classification bias (e.g., complicated shapes have more failure labels) did not appear to hinder convergence. This memory can be prefilled with imitation examples to bootstrap the critic learning. While it is possible to minimize the use of expert knowledge, this paper focuses on the main learning method, while the impact of imitation learning will be analyzed in future work.

## 4.2 MAXIMUM-UNCERTAINTY COLLECTION

Given one state of the environment, the actor can generate several candidate actions. Depending on the training stage and the state, these proposed actions can have a different degree of information for the critic. Selecting actions for which the critic predicts $\xi \approx 0.5$, i.e., it cannot decide between success and failure, can provide higher information content. This strategy has proven to improve the convergence in our tests.

## 4.3 ACTION OPTIMIZATION

Optimal performance in an environment with multiple disconnected sets of feasible actions would require a multimodal distribution with a probability density of zero between the modes. Since the actor is a continuous neural network and the latent space is continuous, the actor cannot generate only positive actions. However, the actor aims to minimize the probability density for actions in the gaps between the modes. The probability density at each action can be estimated using the KDE, and the actions with the lowest corresponding density can be rejected. The accuracy of actors with strong multimodal performance like FKL is significantly increased from action optimization as shown in Section 6.

## 5 EXPERIMENTAL SETUP

### 5.1 ROBOTIC GRASPING SIMULATION

The proposed architecture was tested in a simplified robotic grasping simulation. We assume a vertical configuration of a parallel gripper with three degrees of freedom $x$, $y$, and $\alpha$ and an object that is an extrusion of a 2D shape. The simulator generates five different shapes with varying position, angle, color, and geometric parameters. The success of a grasp is determined by the relative position and alignment of the gripper to the outline of the object as seen from a camera positioned above the experiment. Details about the simulator can be found in Appendix A.

The choice of this simulation, as opposed to existing robotic simulations, was motivated by efficiency while iterating through several combinations of models and parameters. The target distribution fidelity is not of primary concern in this work. The focus is instead on the capability of the proposed method to learn all feasible actions.

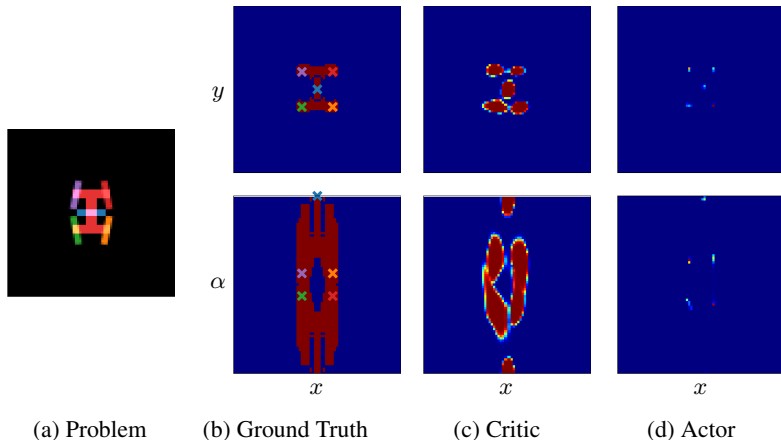

|     (a) Problem     |     (b) Ground Truth     |     (c) Critic     |     (d) Actor     |

Figure 1: Critic classification and actor distribution trained with JS compared with the ground truth. Five example grasps are shown in the problem and their associated locations in the ground truth. The figures show projections onto the x-y plane (top row) and the x-$\alpha$ plane (bottom row).

## 5.2 COMPARISON

In the evaluation, we are comparing different f-divergences with each other and with two other approaches. The analyzed f-divergences are the FKL, RKL, and JS divergences. The two other approaches are an ME RL algorithm similar to SAC in Haarnoja et al. (2018), which trains the actor to minimize

$$\min_{\theta} \mathbb{E}_{s \sim \mathcal{M}, z \sim \mathcal{Z}} \left[ \log q_{\theta}(\pi_{\theta}(s, z)|s) - \xi_{\phi}(s, \pi_{\theta}(s, z)) \right], \tag{12}$$

with $\mathcal{M}$ being the replay memory. The critic is trained as described in Section 4. Instead of using the reparameterization trick with a known distribution to estimate the entropy, we use the KDE. The other approach is an implementation of a conditional GAN (Mirza & Osindero (2014)) with a growing dataset. The min-max optimization problem is given through

$$\min_{\theta} \max_{\phi} \mathbb{E}_{s, a \sim \mathcal{M}_p, z \sim \mathcal{Z}} \left[ \log(\xi_{\phi}(s, a)) - \log(1 - \xi_{\phi}(s, \pi_{\theta}(s, z))) \right], \tag{13}$$

with the positive replay memory $\mathcal{M}_p$, which grows through successful interactions with the environment. An asterisk is added (e.g., JS*) when using action optimization, rejecting 10% of the proposed actions with the lowest probability density. The actor gradient estimators for all approaches are listed in Table 2.

In the following section, we only compare with approaches that do not explicitly utilize the structure of the problem, as the intention of the proposed approach is to be generally applicable in a continuous CB problem setting. A comparison with a widely used approach from the grasping literature is conducted in Appendix D.

## 6 RESULTS

For each configuration, 3 agents were trained for 1 million interaction steps with the environment, taking approximately 48 hours on a single NVIDIA A100 GPU. At the start of the training, 80k examples, including positives and negatives, for randomly generated shapes were added to the balanced replay memory to bootstrap the critic and discriminator learning.

Figure 1 shows the problem, the ground truth feasible picking positions, the critic estimate, and a heat-map of the actor's proposed actions. All figures are projections taking the maximum over the dimension that is not shown. In the problem visualization in Figure 1a, five feasible picks are shown in different colors, which correspond to the markers in Figure 1b. These markers highlight the complex multimodality of the problem. While it appears that, e.g., red and purple are in the same mode in the x-y projection, it is visible in the x-$\alpha$ projection that they are not directly connected. Figure 1c shows that the critic has an approximate understanding of the feasible regions of the action

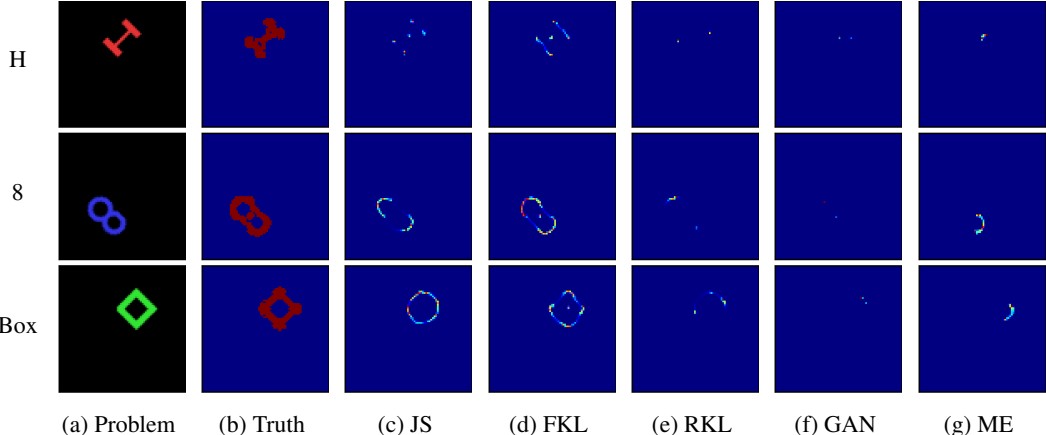

|     |     |     |     |     |     |     |
| :-: | :-: | :-: | :-: | :-: | :-: | :-: |
| (a) Problem | (b) Truth | (c) JS | (d) FKL | (e) RKL | (f) GAN | (g) ME |

Figure 2: Qualitative comparison of the implemented algorithms, showing action heat-maps on three different states, with the last state never been observed during training.

space, showing five modes clearly in the x-y projection. The actor distribution in Figure 1d also shows all five modes, while the output is significantly sharper in the centers of the modes. This is due to the use of the KDEs and the choice of bandwidth $\sigma$.

In the qualitative comparison in Figure 2 the actor distributions of the different algorithms are shown for three different shapes. While the *H* and *8* shapes were trained on, the *Box* shape has not been seen during training. The different subfigures show the action heat maps of all implemented algorithms, showing only the x-y projections. The *H*-row shows that JS and FKL learned all five modes, with JS having the fewest samples in the connecting area. RKL and the GAN show two modes. The ME implementation collapses in a single mode. The *8*-row and the *Box*-row show a similar pattern with the most pronounced spread of the action distributions in JS and FKL and mostly collapsed action regions in the other approaches.

To quantify the multimodal capabilities, and thus the transferability of the learned skill, each algorithm's accuracy and shares of modes on all shapes were evaluated. For each shape, 1024 random states were generated that differ in pose, color, and geometry (example variations can be seen in the Appendix A). For each state, 1024 actions were sampled from the different actors. The actions were then evaluated, and the mode of each action was recorded. The modes were then ranked and averaged over all the states of that shape by frequency. By averaging the ranks instead of the modes, the last rank shows the average ratio of the least frequent mode for each state.

Figure 3 shows the shares of each rank for the *H* and *Box* shapes for all the algorithms, with the asterisk indicating that action optimization was applied. This figure presents the multimodal capabilities of the JS and FKL algorithms, which are the only ones with the last ranked mode present for the *H* and with significantly more pronounced modes than the others for the *Box*. Therefore, only JS and FKL are capable of learning a transferable skill according to our definition. The generalization capability of the GAN implementation is significantly lower than the others, as seen on the *Box* shape.

To quantify the performance, Table 3 shows the accuracy (feasible actions generated over total actions generated) for each shape and the last ranked mode for the *H*, *T*, and *Box* shapes. The table shows that ME has solid performance on all shapes but fails to find the different modes. The GAN algorithm performs well with some modes present, but overall it is weaker than the others. RKL has high scores but mostly fails at finding all the modes. FKL shows good performance in mode finding, with an overall accuracy similar to RKL. JS is on par with the ME accuracy with the addition that it repeatably finds all the modes. Generally, action optimization improves accuracy but does not help mode finding, slightly decreasing the least ranked mode for most approaches. The maximum deviations in the superscript show that all approaches learn reliably with the GAN having the highest performance deviations among runs.

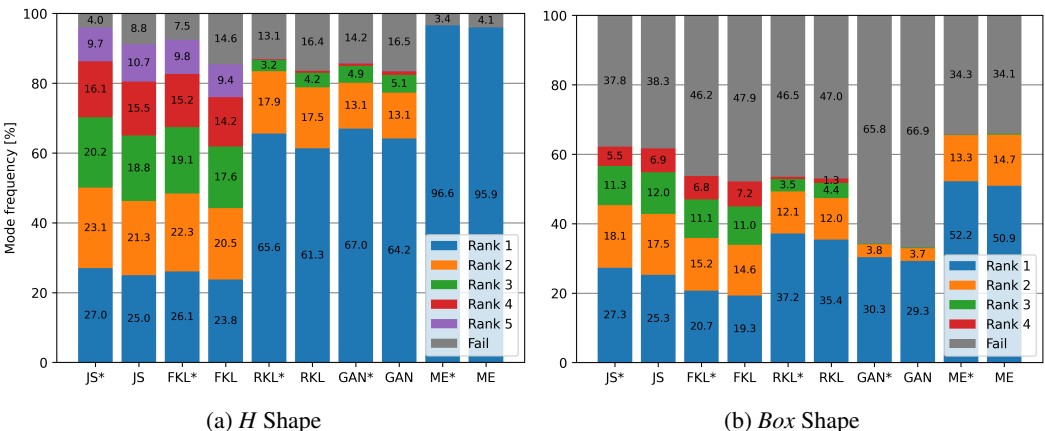

(a) *H* Shape  (b) *Box* Shape

Figure 3: Gripping rank comparison, with the ratio of picks for each ranked mode or failure in %.

| | | JS* | JS | FKL* | FKL | RKL* | RKL | GAN* | GAN | ME* | ME |
|---|---|---|---|---|---|---|---|---|---|---|---|
| Score | H | $96.0^{1.0}$ | $91.2^{1.3}$ | $92.5^{0.4}$ | $85.4^{1.1}$ | $86.9^{3.6}$ | $83.6^{3.5}$ | $85.8^{2.0}$ | $83.5^{2.7}$ | $\mathbf{96.6}^{0.7}$ | $95.9^{1.2}$ |
| | T | $96.5^{0.5}$ | $93.3^{1.0}$ | $92.5^{1.0}$ | $88.0^{0.9}$ | $\mathbf{97.6}^{0.9}$ | $95.7^{1.2}$ | $81.6^{2.5}$ | $79.7^{1.7}$ | $93.9^{1.9}$ | $93.6^{1.7}$ |
| | 8 | $\mathbf{86.9}^{2.3}$ | $82.9^{1.7}$ | $83.4^{0.7}$ | $77.7^{0.7}$ | $82.8^{3.2}$ | $79.2^{2.5}$ | $70.3^{4.5}$ | $66.9^{5.7}$ | $85.5^{0.5}$ | $84.2^{1.9}$ |
| | Spoon | $97.4^{0.6}$ | $96.9^{0.5}$ | $93.6^{1.9}$ | $90.6^{2.3}$ | $97.4^{1.1}$ | $\mathbf{97.6}^{1.1}$ | $94.7^{2.1}$ | $95.1^{2.3}$ | $92.1^{1.5}$ | $92.2^{1.6}$ |
| | Box | $62.2^{2.3}$ | $61.7^{2.7}$ | $53.8^{7.4}$ | $52.1^{7.1}$ | $53.5^{2.9}$ | $53.0^{3.4}$ | $34.2^{8.2}$ | $33.1^{7.2}$ | $65.7^{6.9}$ | $\mathbf{65.9}^{6.8}$ |
| | Avg | $\mathbf{87.8}^{0.8}$ | $85.2^{0.8}$ | $83.2^{0.7}$ | $78.8^{0.6}$ | $83.6^{2.1}$ | $81.8^{2.1}$ | $73.3^{2.3}$ | $71.7^{2.7}$ | $86.8^{2.1}$ | $86.4^{1.8}$ |
| Mode | H | $9.7^{0.7}$ | $\mathbf{10.7}^{0.3}$ | $9.8^{0.5}$ | $9.4^{0.6}$ | $0.0^{0.0}$ | $0.0^{0.0}$ | $0.2^{0.4}$ | $0.2^{0.3}$ | $0.0^{0.0}$ | $0.0^{0.0}$ |
| | T | $11.0^{0.7}$ | $12.3^{0.6}$ | $\mathbf{15.8}^{0.6}$ | $15.3^{0.3}$ | $0.4^{0.3}$ | $0.7^{0.4}$ | $1.1^{2.3}$ | $1.4^{2.6}$ | $0.0^{0.0}$ | $0.0^{0.0}$ |
| | Box | $5.5^{0.8}$ | $6.9^{1.1}$ | $6.8^{1.2}$ | $\mathbf{7.2}^{1.4}$ | $0.7^{0.8}$ | $1.3^{1.3}$ | $0.0^{0.0}$ | $0.0^{0.0}$ | $0.0^{0.0}$ | $0.0^{0.0}$ |

Table 3: Comparison on all shapes with the mean of the grasping success ratio in % on top and the least ranked mode in % on the bottom, with the maximum deviations over the 3 runs in superscript.

## 7 CONCLUSION AND FUTURE WORK

This work proposes to learn a skill by training a generator to generate all feasible actions for a subtask. To this end, the output distribution of the generator is learned to match a uniform distribution over the feasible action space. While learning within a 2D grasping simulation, the method shows stable convergence for FKL, RKL, and JS. As expected, FKL is more efficient in visiting all the modes. JS has the highest accuracy while reliably visiting all the modes. The proposed learning strategy expands the current state-of-the-art training within multimodal interactive environments by showing competitive accuracy while visiting all the modes. Since the proposed approach can visit all the modes, it learns the skill of grasping independently of a specific downstream task. In future work, we will investigate how downstream tasks can choose their optimal action among the proposed feasible options. As it is not dependent on the structure of the problem, we will further utilize it for a 6D grasping problem as well as for other applications.

Some limitations have emerged during experimentation. Currently, many imitation examples are required to bootstrap the critic's learning. A possibility to mitigate this could be the progressive tuning of the KDEs or learning their parameters during training. This approach could favor exploration initially and divergence estimation later in training. A complementary strategy could be using curriculum learning techniques that start with simple problems where solutions are less sparse in the action space. Furthermore, the proposed approach may not be very effective in high-dimensional problems as the sampling requirements for the estimator grow exponentially. The limit on the degrees of freedom will be explored in future work. A mitigation of this issue can come from the rich literature on non-parametric density estimation in higher-dimensional problems and its applicability to the proposed method. Another approach could be to split higher-dimensional problems into multi-step low dimensional problems and to learn to generate all feasible trajectories.

## 8 REPRODUCIBILITY STATEMENT

The final version will include a GitHub link to the code for learning and evaluation.

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

## A  GRASPING SIMULATION

The grasping simulator generates four different shapes (H, 8, Spoon, T) for training and a Box shape for testing. The shape position, orientation, color, and geometry parameters are randomly sampled, producing various observations. The observation space is a $128 \times 128$ pixel RGB image. We assume a vertical configuration of a parallel gripper with three degrees of freedom $x$, $y$, and $\alpha$ and assume that the object is an extrusion of the 2D shape in the observation. The action space is constrained to the center $78 \times 78$ pixel region to avoid undefined behavior at the border of the RGB image. The angle of the grasp is in $[0, \pi)$ as the gripper is symmetrical, and thus a full revolution is not necessary.

The success of a grasp is only determined by the relative position and alignment of the gripper to the outline of the object, as seen from a camera positioned above the experiment. We developed an algorithm that, given the alignment of the gripper, i.e., $x$, $y$, and $\alpha$ and a simulated picture of the object from a fixed camera, provides a success/failure outcome in a deterministic and reproducible manner. Given the maximum aperture of the parallel gripper $l$ and the width of the gripper claws $w$, the simulation analyzes the cropped image content of dimensions $l \times w$ between the gripper claws before the claws close on the object. The simulation checks if the object is sufficiently present and equidistant from the claws and aligned within parameterized margins. Figure 4 shows successful grasping poses and the respective gripper content for the objects that are trained on.

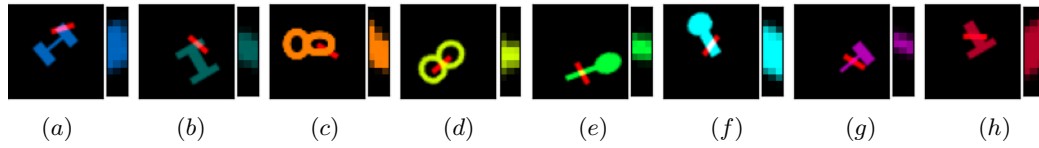

(a)          (b)          (c)          (d)          (e)          (f)          (g)          (h)

Figure 4: Feasible gripper positions (red) for different variations of the shapes (*H-shape*, *8-shape*, *Spoon*, and *T-shape*) used in training, with a detailed view of the area between the gripper to the right of each figure.

## B  NEURAL NETWORK ARCHITECTURES

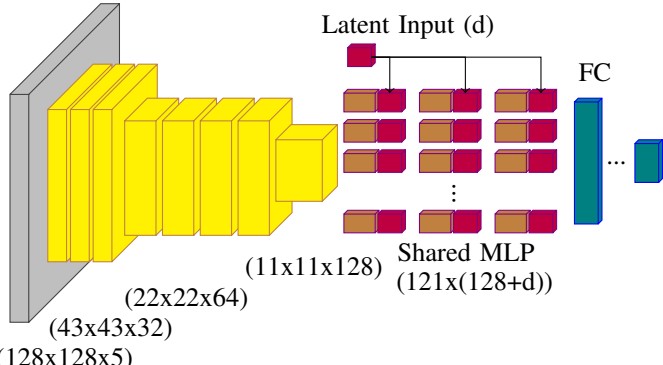

Figure 5: Before processing, the image is augmented with positional encoding resulting in 5 total channels $\{r, g, b, x, y\}$. The network's input layer (in gray) is a 5x5 embedding layer with stride 3, followed by 7 residual blocks (in yellow) with a bottleneck. The output is processed by 3 layers of "pixelwise" shared MLPs (in brown), with the features being concatenated with a latent input (in purple) of length $d$. The latent input is a random sample from $\mathcal{Z}$ for the actor and the action to be evaluated for the critic. Four (for the actor) or three (for the critic) fully connected layers (in blue) output the action and the score, respectively.

The neural network design was guided by simplicity and inspired by GANs. Features that rely on domain-specific knowledge are avoided to evaluate better the learning method presented in the paper.

The structure of the actor and critic neural networks are illustrated in Figure 5. The residual feature extraction network (He et al. (2016)) is shared between the actor and critic.

As a peculiarity of the network and the loss, the actor's inferred action has four components, $[x, y, r\sin\alpha, r\cos\alpha]$, with $r \in [0, \sqrt{2}]$. The angle can be extracted trivially with the $\arctan$ of the ratio of the third and fourth action components. As the scale factor $r$ does not change the angle, the critic receives the normalized action $[x, y, \sin\alpha, \cos\alpha]$ as input. To avoid the actor from reaching the singularity at $r = 0$ and the distribution $q$ being spread along the radius, $g(s, a)$ and $\xi(s, a)$ are scaled with an unnormalized Gaussian on the radius, centered at $0.5$ with the standard deviation of $0.4$.

## C   ALGORITHM AND HYPERPARAMETERS

---

**Algorithm 1:** Jenson-Shannon training loop

---

1   Initialize $\mathcal{M}$ with imitation and random examples and initialize $\theta, \phi$

2   **for** 1 **to** *Training Steps* **do**    `// Training steps are 1,000,000 in experiments`

3     **for** 1 **to** *Interaction Steps* **do**   `// Interaction steps are 1 in experiments`

4       $s \leftarrow$ Generate a new problem

5       $z_i \leftarrow$ Sample uniformly in $\mathcal{Z}, \quad \forall i \in [1, U]$

6       $a_i \leftarrow \pi_\theta(s, z_i), \quad \forall i \in [1, U]$

7       $\hat{r}_i \leftarrow \xi_\phi(s, a_i), \quad \forall i \in [1, U]$

8       $j \leftarrow \arg\min_{i \in [1, U]} |0.5 - \hat{r}_i|$   `// Get action with highest uncertainty`

9       $r \leftarrow g(s, a_j)$

10       **if** *r == 1* **then**

11        Store $(s, a_j, r)$ in $\mathcal{M}_p$

12       **else**

13        Store $(s, a_j, r)$ in $\mathcal{M}_n$

14       **end**

15     **end**

16     **for** 1 **to** *Critic Steps* **do**       `// Critic steps are 2 in experiments`

17       $(s_i, a_i, r_i)_{i=1}^L \leftarrow$ Sample half from $\mathcal{M}_p$ and half from $\mathcal{M}_n$

18       $\phi \leftarrow \phi - \alpha_\phi \nabla_\phi \sum_{i=1}^L r_i \log(\xi_\phi(s_i, a_i)) + (1 - r_i)\log(1 - \xi_\phi(s_i, a_i))$

19     **end**

20     **for** 1 **to** *Actor Steps* **do**       `// Actor steps are 1 in experiments`

21       **for** $k = 1$ **to** $K$ **do**

22        $s_k \leftarrow$ Sample from $\mathcal{M}$

23        $z_i \leftarrow$ Sample uniformly in $\mathcal{Z}, \quad \forall i \in [1, N]$

24        $a_i \leftarrow \pi_\theta(s_k, z_i), \quad \forall i \in [1, N]$

25        $\epsilon_j \sim \mathcal{N}(0, \sigma'), \quad \forall j \in [1, M]$

26        $a_j^* \leftarrow \text{stop\_gradient}(a_{\lceil j/m \rceil}) + \epsilon_j, \quad \forall j \in [1, M]$   `// Resample from KDE`

27        $\hat{q}_j \leftarrow \frac{1}{N}\sum_{i=1}^N k_\sigma(a_j^* - a_i), \quad \forall j \in [1, M]$   `// Evaluate KDE on samples`

28        $\hat{q}_j' \leftarrow \frac{1}{N}\sum_{i=1}^N k_{\sigma'}(a_j^* - a_i), \quad \forall j \in [1, M]$    `// Evaluate proposal pdf`

29        $\hat{r}_j \leftarrow \xi_\phi(s_k, a_j^*), \quad \forall j \in [1, M]$

30        $\hat{V} \leftarrow \frac{1}{M}\sum_{j=1}^M \frac{\hat{r}_j}{\hat{q}_j'}$   `// MC integration with importance sampling`

31        $\hat{p}_j \leftarrow \frac{\hat{r}_j}{\hat{V}}, \quad \forall j \in [1, M]$

32        $g_k \leftarrow \frac{1}{2M}\sum_{j=1}^M \frac{\hat{q}_j}{\hat{q}_j'}\log\left(\frac{2\hat{q}_j}{\hat{q}_j + \hat{p}_j}\right)\nabla_\theta \log(\hat{q}_j)$       `// gradient trace`

33       **end**

34       $\theta \leftarrow \theta - \alpha_\theta \frac{1}{K}\sum_{k=1}^K g_k$

35     **end**

36   **end**

---

| Parameter | Value | Description |
|---|---|---|
| N | 128 | Minibatch size |
| M | 256 | Resampling size |
| m | 2 | Samples per KDE support point ($M/N$) |
| U | 64 | Maximum uncertainty proposals |
| K | 16 | Actor batch size |
| L | 32 | Critic batch size |
| $\sigma$ | diag(0.025, 0.025, 0.4, 0.4) | KDE bandwidth |
| $\sigma'$ | diag(0.075, 0.075, 1.2, 1.2) | Sampling KDE bandwidth |
| $|\mathcal{M}_p|$ | 160,000 | Positive replay memory size |
| $|\mathcal{M}_n|$ | 160,000 | Negative replay memory size |
| $|\mathcal{M}|$ | 320,000 | Total replay memory size |

Table 4: Hyperparameters

# D OBSERVATION VARIATION EXPERIMENTS

## D.1 SETUP

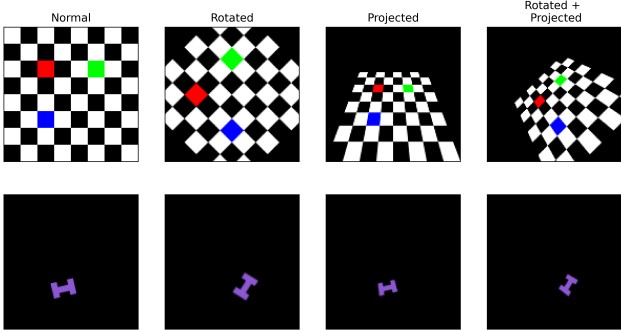

Figure 6: Different distortions are applied, showing a colored chess board for illustration and an example shape under all distortions.

To highlight the difference between the proposed approach and related work of robotic grasping, we investigate how distortions of the observation affect the performance. The distortions investigated are a rotation, projection, and rotation + projection as shown in Figure 6. These distortions correspond to different camera perspectives. We train a new agent for $10^6$ training steps for each distortion and approach in the following comparison.

We compare with a common approach in the literature (Zeng et al. (2020)) that make use of spatial equivariances. The approach utilizes fully convolutional networks to output a probability of success for each action of a discretized action space. We implement two variants. In the first one, just as in Zeng et al. (2020), the observation is fed into the neural network multiple times with different rotations. The neural network then only needs to output a one-channel image containing the probability of success of each discretized $x, y$ action for the given rotation of the image. This approach thus makes use of translation equivariance by using a convolutional neural network (CNN) and rotation equivariance. In the experiments, we denote it as the heat-map approach (H). The second variant estimates for each observation the success for different rotations by outputting a multi-channel image indicating the success estimate of each discretized $x, y, \alpha$ action explicitly. Thus it only takes advantage of translation equivariance. It is called stacked heat-map (SH) in the following.

The approaches are implemented using fully convolutional networks with an hourglass structure, adopting the beginning of the Resnet in Figure 5 and adding the same structure in reverse order with nearest-neighbor upsampling. Both approaches predict grasping success for 78x78 pixels with 16 rotation angles. They are trained on a cross-entropy loss on the grasping outcome sampled from the balanced replay buffer. The replay buffer is also filled with imitation learning examples, and maximum uncertainty sampling is applied. For evaluation, the success estimate of each discretized

action is used as its probability to be sampled. To increase accuracy, an inverted temperature factor increases the difference between higher and lower score actions. Specifically, the actions are sampled according to

$$q(a|s) = \frac{\exp(\beta \log \xi(s,a))}{\sum_{\forall a \in \mathcal{A}_d} \exp(\beta \log \xi(s,a))}, \tag{14}$$

with $\xi$ being the fully convolutional network with $s$ as input and as output shape the discretized action space $\mathcal{A}_d$. The inverted temperature was set to $\beta = 100$.

## D.2 RESULTS

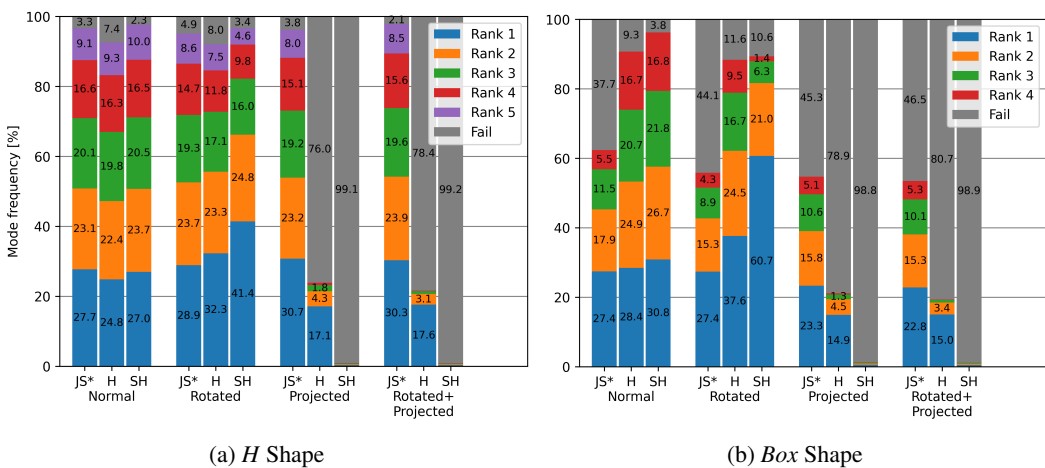

(a) *H* Shape

(b) *Box* Shape

Figure 7: Gripping rank comparison, with the ratio of picks for each ranked mode or failure in %.

|  |  | Normal | | | Rotated | | | Projected | | | Rotated + Projected | | |
|---|---|---|---|---|---|---|---|---|---|---|---|---|---|
|  |  | JS* | H | SH | JS* | H | SH | JS* | H | SH | JS* | H | SH |
| Score | H | 96.7 | 92.6 | **97.7** | 95.1 | 92.0 | **96.6** | **96.2** | 24.0 | 0.9 | **97.9** | 21.6 | 0.8 |
|  | T | 97.1 | 93.5 | **98.2** | 95.5 | 94.3 | **98.3** | **96.2** | 31.6 | 0.8 | **98.0** | 27.0 | 0.7 |
|  | 8 | 86.4 | 90.4 | **97.2** | 85.5 | 87.4 | **95.5** | **85.3** | 15.4 | 0.9 | **88.8** | 14.3 | 0.8 |
|  | Spoon | 97.7 | 93.8 | **98.8** | 96.5 | 94.5 | **98.6** | **97.1** | 33.8 | 0.7 | **97.8** | 33.2 | 0.6 |
|  | Box | 62.3 | 90.7 | **96.2** | 55.9 | 88.4 | **89.4** | **54.7** | 21.1 | 1.2 | **53.5** | 19.3 | 1.1 |
|  | Avg | 88.0 | 92.2 | **97.6** | 85.7 | 91.3 | **95.7** | **85.9** | 25.2 | 0.9 | **87.2** | 23.1 | 0.8 |
| Mode | H | 9.1 | 9.3 | **10.0** | **8.6** | 7.5 | 4.6 | **8.0** | 0.2 | 0.0 | **8.5** | 0.0 | 0.0 |
|  | T | 11.5 | **18.7** | 18.5 | 10.4 | **14.8** | 9.8 | **11.8** | 0.5 | 0.1 | **14.3** | 0.2 | 0.1 |
|  | Box | 5.5 | 16.7 | **16.8** | 4.3 | **9.5** | 1.4 | **5.1** | 0.3 | 0.1 | **5.3** | 0.1 | 0.1 |

Table 5: Comparison of the proposed Jensen-Shannon approach with approaches from the literature.

The results are shown in Figure 7 and Table 5, which compare the performance of the proposed Jensen-Shannon (JS) approach with the heat-map (H) and stacked heat-map (HS) approaches on the four different distortions. As expected, the specifically crafted H and SH approaches perform better on the original problem than the generic approach proposed in the paper. In that scenario, no scene understanding is required, and only local features need to be considered to estimate grasping success. Therefore, the approaches are expected to generalize well to unseen shapes, as seen for the Box-Shape, since the grasping success depends only on gripper alignment. They only need to learn to imitate the grasping success heuristic shown in Figure 4.

Interestingly, rotating the observation does not seem to impact their performance. However, under projection and projection + rotation, both approaches fail to learn to grasp. The heat-map approach

still learns to grasp the objects occasionally, while the stacked heat-map approach does not learn anything. Our proposed approach learns well for all distortions. The generalization on the Box-Shape decreases a bit but the general performance remains similar. Surprisingly, the results for rotation + projection on the training shapes are above the variance in the main results in Table 3. An explanation for the performance increase could be that the region in the input observation that the object can be in is smaller than in the normal case, as can be seen by the area between the colored tiles in Figure 6. The reduced region could slightly ease scene understanding, leading to improved results.

In general, the performance of our proposed approach does not depend on the distortion as it does not explicitly use the spatial structure. Its design does not depend on the specifics of the experiment at all. It can, therefore, learn independently of the distortion applied as long as the object is still fully observable.

