# OpenReview forum: "Learning to Generate All Feasible Actions"
_ICLR.cc/2023/Conference — Submitted to ICLR 2023_

### Official Review · Reviewer_RMrn · 2022-10-21

**Confidence:** 3
**Correctness:** 3
**Technical Novelty And Significance:** 4
**Empirical Novelty And Significance:** 4
**Recommendation:** 8

**Clarity, Quality, Novelty And Reproducibility:**

The work seems original and interesting, and after more development, could be broadly applicable.

Questions:

(1) In Section 4.2, do you choose your high-uncertainty actions probabilistically, or take the one with the smallest margin?  On a related note, this approach sounds a lot like an active learning problem; could you substitute a different AL algorithm here?

(2) When you contrast JS/FKL to ME, you note that they have similar failure rates but more diverse outputs from JS/FKL.  What kind of tradeoff do you think is acceptable between failure rate and diversity?  I.e., how much higher of a failure rate (over ME) would you accept to claim that the higher diversity of outputs worthwhile?

(3) The idea of generating all actions sounds a little like the idea of "diverse density" from multiple-instance learning.  Are there any results from that family of algorithms that might be useful here?


**Strength And Weaknesses:**

Strengths:
 - The idea of generating all feasible actions is an interesting one, and has intuitive appeal to facilitate skill transfer.
 - The authors present novel approaches to estimate gradients in various sampling contexts.
 - The authors present novel approaches to reduce bias in training the critic.
 - Empirical results are encouraging.

Weaknesses:
 - Typo on page 8: "RKL, has" -> "RKL has"
 - The legend of Figure 3(b) covers some numbers


**Summary Of The Paper:**

The authors address the problem of large solution spaces by decomposing problems into smaller subproblems and training agents to consider all feasible actions for the subproblems, hopefully enhancing transferability of solutions to new problems.  They develop a generative model to produce all feasible actions and evaluate it on a 2D grasping problem.


**Summary Of The Review:**

I think that this is a paper worthy of publication in ICLR, though my confidence is not high.  It clearly defines a problem and concisely presents their approaches.

---

> ### Author Response · Authors · 2022-11-18
> **Answer to the reviewer comments**
>
> We would like to thank the reviewer for the helpful comments and questions. We have fixed the typo on page 8 and placed the legend in Figure 3b correctly. We uploaded a revision with changes highlighted. Regarding your questions:
>
> 1. The high-uncertainty actions are chosen according to the smallest margin, i.e., the one closest to 0.5 is selected (Algorithm 1, Line 8 in Appendix C). Active learning could indeed be an interesting direction to improve sample efficiency and we plan to test applicability of AL algorithms in the future.
>
> 2. We expanded on the idea of the skill in the revision. To summarize, for a skill to be reusable all modes need to be visited. Otherwise, some downstream tasks may be impossible because the necessary grasp is in a mode that is not learned. Therefore, learning all modes is essential for learning a skill and therefore as important as overall accuracy.
>
> 3. We would like to thank the reviewer for pointing us towards MIL. So far, we did not utilize any approaches such as diverse density but they could indeed offer valuable inspiration for improving our algorithm.
>
> We would like to invite the reviewer to inspect the changes made to the paper.

---

### Official Review · Reviewer_bb3F · 2022-10-24

**Confidence:** 2
**Correctness:** 3
**Technical Novelty And Significance:** 3
**Empirical Novelty And Significance:** 2
**Recommendation:** 5

**Clarity, Quality, Novelty And Reproducibility:**

I'm unsure of the novelty of the approach, lots of works have studied exploration ideas in RL, but this is not an area I am too familiar with.

The writing is well done and explains the idea and method well. I'm not sure if it will be reproducible from the paper, as it is heavy on equations but has no implementation details or experimental setup explanation. The paper states a github link will be given, but at the moment, only the paper is available.

**Strength And Weaknesses:**

The paper works on an interesting and important problem of finding/exploring all actions. The paper is well written and motivates the problem nicely.

The biggest weakness is the lack of experiments. There is only one experiment, and its setup, environment, action space, etc. are not defined. Further, there is no comparison to other methods, other than a few proposed in the paper. This makes it difficult to judge the work and its contribution.

It is also unclear how much the proposed approach helps vs other things used (imitation learning, etc.)

**Summary Of The Paper:**

This paper introduces an RL method designed to generate all possible actions. I assume this is done to help exploration of the space in order to learn a better policy. The approach is evaluated on a grasping task.

**Summary Of The Review:**

The paper proposes a method to generate all possible actions and is evaluated on a grasping setup. The paper is well written, but the experiments are weak/minimal.

---

> ### Author Response · Authors · 2022-11-18
> **Answer to the reviewer comments**
>
> We would like to thank the reviewer for the feedback. We hope we were able to address the comments in our revision:
>
> - We described the simulator in more detail, adding a description of the observation and action spaces. The detailed description can be found in Appendix A.
> - We provided a description of the related grasping work and highlighted why our algorithm design is necessary. The discussion can be found at the end of the related work section.
> - Additionally, we added a variation of the experiment where we compare our method to related work and show that it does not require carefully crafted experimental setups as the related work. We distorted the observation with a projection, which corresponds to using camera positions that are not perfectly top-down. The results can be seen in Appendix E. We invite the reviewer to inspect the new experiments and hope we were able to address the concerns.
> - Since we cannot publish the github link before the review is finalized we added a full description of the algorithm in Appendix C.
>
> We invite the reviewer to inspect our changes (highlighted in the revision) and hope that they address the reviewer’s concerns.

---

> > ### Comment · Reviewer_bb3F · 2022-11-30
> > **response**
> >
> > The rebuttal and revised version have addressed many of my concerns. The main remaining concern is that there still aren't comparisons to other works, which makes it hard to understand how it performs and how useful it would be.

---

> > > ### Author Response · Authors · 2022-12-01
> > > **Comparison**
> > >
> > > We added a comparison with heatmap approaches that are common in the literature in Appendix D. It was intended to show the limitation of these approaches, which is explicit dependence on spatial equivariances, and highlight the independence of our proposed approach.

---

### Official Review · Reviewer_JUVt · 2022-10-24

**Confidence:** 3
**Clarity, Quality, Novelty And Reproducibility:** The proposed method is novel. The the…
**Correctness:** 3
**Technical Novelty And Significance:** 3
**Empirical Novelty And Significance:** 3
**Recommendation:** 6

**Strength And Weaknesses:**

The problem considered in this paper is to generate all feasible actions for a certain task which is hard to obtain for complex tasks. The author propose a novel learning method for generative neural network in order to generate feasible actions. In order to solve the problem, the authors further propose a gradient estimator for the gradient of f-divergence. The proofs are rigorous and correct.

Weakness: I found this paper a bit hard to follow since some theories are not well explained in the paper. It would be great if the authors can provide more comprehensive explanation for the result you used in the paper.

**Summary Of The Paper:**

This paper proposes a method to train a generative neural network to generate all feasible actions of a sub task by interacting with an environment. The trained generator provides a distribution that matches a uniform distribution over all feasible actions. The authors derive the general optimization target for arbitrary f-divergences using a combination of kernel density estimates, resampling, and importance sampling. An application of a 2D robotic grasping problem is presented in this experiment to show the proposed algorithm outperforms other related methods.

**Summary Of The Review:**

This paper considers a very important problem in real application. The proposed method is novel and well supported by theoretical and experimental results.

---

> ### Author Response · Authors · 2022-11-18
> **Answer to the reviewer comments**
>
> We would like to thank the reviewer for the review. In the revision, we aimed at providing more details and more context to improve comprehensiveness. The following descriptions were added:
>
> - Comparison to related grasping literature (Section 2)
> - Detailed description of the simulator (Appendix A)
> - Algorithm of the Jensen-Shannon approach (Appendix C)
> - Additional experiments (Appendix E)
> - Where needed, we added more details in various sections
>
> We invite the reviewer to inspect our changes (highlighted in the revision) and hope that they improve the readability of our work.

---

### Official Review · Reviewer_fUjg · 2022-10-25

**Confidence:** 4
**Correctness:** 4
**Technical Novelty And Significance:** 4
**Empirical Novelty And Significance:** 3
**Recommendation:** 3

**Clarity, Quality, Novelty And Reproducibility:**

Quality and novelty are high as stated above.
The paper as it is is lacking in clarity of motivation and terminology as well as comparisons.
The method is clearly explained with enough details to allow for reproducibility.

**Strength And Weaknesses:**

The strengths of the paper are:
1. It is theoretically well motivated and sound. The authors explore various f-divergences to find the best one for this setting.
2. The analysis of the various modes and grasping success for different methods is thorough for the chosen domain.
3. As far as I can tell, their approach is novel.

The two biggest areas for improvement are:
1. Lack of comparison with any relevant literature - grasping is well studied problem and comparison with some method in literature besides author-crafted baselines is required. The authors explore a few variations of their own method as well as compare to some sanity-check baselines but  the grasping success % can be compared to other methods in literature. I also found the related works section lacking in references to grasping literature and instead referencing peripheral works such as LLMs.
2. Unclear writing - It is unclear which problem the method is solving, that of generating feasible actions or discovering skills or both. Skills are typically referring to a sequence of actions whereas the case considered here is closer to CB with a single state/action. It seems the terms "multimodal action distributions" and "skills" are being conflated and further clarification is needed. Moreover, while in the abstract the claim is to learn "reusable and transferrable skills", there is no discussion later in the paper about the reusability or transferability of the action modes discovered.

Smaller points of feedback:

3. It is unclear what happens in environments where the hand-designed heuristic for judging success/failure is not present. The method is only tested on a single environment of 2D grasping and could benefit from being tested on at least one other problem to get an idea of how well it works as a general method for feasible action generation.
4. The role of the "critic" network is not clearly explained. Why is it that the learned critic can generalize over environment samples better, and hence provide better gradients, than directly training the KDE on them?
5. What is the role of the imitation learning dataset used to bootstrap the critic? An ablation on how much data is needed there would be interesting to see.
6. A study of the sample complexity in terms of environment interactions would be a relevant result.
7. Results for the figure 8 shape are missing from figure 3.



**Summary Of The Paper:**

This paper presents an approach to discover the feasible actions for a given state in an environment. The algorithm proposes actions, which end in a single step in either success or failure, as determined here by a hand-designed heuristic. The generated actions are multimodal in nature and show limited generalization to out-of-distribution tasks. The method is tested on a robotic gripping task with 3-4 shapes. The results demonstrate that, in this domain, the proposed method can discover more modes of the feasible action distribution than the baselines.

**Summary Of The Review:**

This is a novel and sound method for multimodal action discovery and with proper comparison, clearer writing, and demonstrating the method's generality on at least one other domain it can be a strong contribution.

---

> ### Author Response · Authors · 2022-11-18
> **Answer to the reviewer comments**
>
> We would like to thank the reviewer for the valuable feedback. We have addressed the questions and submitted a revision with changes (highlighted in the revision) that we want to invite the reviewer to inspect. Our answers to the comments are the following:
>
> 1. While we did not include the robotic grasping literature in the initial submission, because we considered the grasping problem as only an example application, we understand how this can be relevant for the context of the proposed work. Therefore, we added a paragraph to the related work section that discusses the grasping literature. We also removed general ML literature (NLP, CV) that was not relevant for the specific context of this work.
> Furthermore, we added a comparison to Appendix D. The specifically crafted approaches of the literature perform better on the original problem than our generic approach. However, under projection, the other approaches fail to learn to grasp, while our approach learns unchanged. That is because the approach is not specifically designed for one setup, but independent of the structure of the experiment.
>
> 2. We removed the term “multimodal action distribution” and elaborated further on our definition of skill, which is: “we consider a skill to be defined by the set of all feasible actions of a subtask.” We added an explanation of a way to measure the learning of a skill as visiting all the grasping modes and we refer to it throughout the work where relevant.
>
> 3. Exploring the application of the approach to different domains as well as different experiments in robotic grasping would indeed be relevant. While we were not able to provide experimentation in a different domain or with a different simulation engine, we hope that the added experiment with varying camera perspectives shows the potential of our proposed approach. In future work, we will apply the approach to different domains.
>
> 4. The role of the critic is to allow the actor to test multiple actions on the same problem in order to compute the actor gradients (see Algorithm 1. Line 29). This is not possible if the environment is a physical experiment or computationally prohibitive if the simulation is complex. We added a sentence in the beginning of Section 4 that should clarify why the critic is needed.
>
> 5.+6. A study of the need for imitation data and the sample complexity would indeed be interesting. There are a few parameters in Algorithm 1 in the Appendix that could significantly improve sample complexity, such as the critic and actor training steps per environment interaction step. As this paper is focusing on establishing the method we did not yet aim at reducing the imitation and environment interactions but we believe that there exists significant potential for improvement.
>
> 7. The 8-Shape has 2-4 modes depending on the width and length of its circles. In some configurations the 8 can be grasped at the center with the gripper in the holes or even with the gripper outside. Since the number of modes is not fixed, we chose not to use this shape for mode rank comparison.

---

> > ### Comment · Reviewer_fUjg · 2022-12-01
> > **Thanks**
> >
> > Thanks to the authors for the clarifications. The related works and experimental setup have been explained better. An additional experiment with rotation+projection of the plane on which the object rests has also been done. I think the paper is stronger than previously overall.
> >
> > My concern that this is a method specific to grasping remains. It is not clear if it will generalize to any other domain as experiments have not been done. As it stands, I cannot recommend acceptance of the paper as it is presented as a general method but it cannot be proven that the many components introduced only help in this specific setting.

---

### Decision · Program_Chairs · 2023-01-20

**Decision:**

Reject

**Justification For Why Not Higher Score:**

The paper is strong overall, with clear novelty on the proposed approach. However, the remaining concern is an important one - the authors claim that their method is generally applicable, but only evaluate on a single domain. What is more, the evaluation setup includes a hand-crafted success heuristic, meaning that the authors would need to somehow show how easily similar results can be achieved in other domains. I could see a version of this paper that makes less general claims and instead focuses purely on grasping, in which all claims are supported. However, this version might be more suitable for a domain specific (e.g., robotics) conference.

**Justification For Why Not Lower Score:**

N/A

**Metareview: Summary, Strengths And Weaknesses:**

The paper addresses the problem of learning all feasible actions (i.e., covering all modes) in interactive settings with complex action spaces. The authors propose a generative approach trained using a general optimization target for f-divergences derived by the authors. The approach is evaluated on a 2D grasping task.

Reviewers found the problem tackled here well motivated and important. A novel, systematic approach is proposed, based on rigorous and correct proofs, and shows encouraging results in a grasping problem domain. Thorough analysis of the modes and success in this target domain complete the picture.

Initial reviews pointed out some weaknesses, some of which have been addressed by the authors during the rebuttal period. Reviewers noted a lack of discussion of related works specific to grasping, some lack of clarity about the problem setting and concepts such as "skills", and some aspects of the provided theory. Reviewers are satisfied with how these points were addressed.

The single biggest shortcoming noted by reviewers is the focus on a single domain for empirical validation. While compelling and important, the grasping problem studied here is just a single example, and does not go far enough to support the claim that this proposed method is generally applicable. A particular concern are hand-crafted components such as the grasping success heuristic - how easily could the overall approach be translated to other settings?

The final recommendation for this paper is taking into account discussion between reviewers. While all reviewers agree that this is high-quality work with novel contributions and potential for impact, they are uncomfortable recommending acceptance given the remaining concern about the evaluation setup and generality of the proposed approach. Therefore, the recommendation is not to accept the paper at the present stage.